# Rapid Indentification of Auramine O Dyeing Adulteration in *Dendrobium officinale*, *Saffron* and *Curcuma* by SERS Raman Spectroscopy Combined with SSA-BP Neural Networks Model

**DOI:** 10.3390/foods12224124

**Published:** 2023-11-14

**Authors:** Leilei Zhang, Caihong Zhang, Wenxuan Li, Liang Li, Peng Zhang, Cheng Zhu, Yanfei Ding, Hongwei Sun

**Affiliations:** 1Key Laboratory of Specialty Agri-Products Quality and Hazard Controlling Technology of Zhejiang Province, College of Life Sciences, China Jiliang University, Hangzhou 310018, China; zhangleilei@cjlu.edu.cn (L.Z.); zchyfdcy@163.com (C.Z.); 18758062145@163.com (W.L.); pzhch@cjlu.edu.cn (C.Z.); 2Agricultural Technology and Soil Fertilizer General Station, Garze Tibetan Autonomous Prefecture, Kangding 626000, China; liliang2836100@163.com (L.L.); 15808368811@163.com (P.Z.); 3School of Automation, Hangzhou Dianzi University, Hangzhou 310083, China

**Keywords:** surface-enhanced Raman spectroscopy, Chinese herbal medicine, Auramine O, dyeing adulteration, machine learning

## Abstract

(1) Background: Rapid and accurate determination of the content of the chemical dye Auramine O(AO) in traditional Chinese medicines (TCMs) is critical for controlling the quality of TCMs. (2) Methods: Firstly, various models were developed to detect AO content in *Dendrobium officinale* (*D. officinale*). Then, the detection of AO content in *Saffron* and *Curcuma* using the *D. officinale* training set as a calibration model. Finally, *Saffron* and *Curcuma* samples were added to the training set of *D. officinale* to predict the AO content in *Saffron* and *Curcuma* using secondary wavelength screening. (3) Results: The results show that the sparrow search algorithm (SSA)-backpropagation (BP) neural network (SSA-BP) model can accurately predict AO content in *D. officinale*, with *R_p_*^2^ = 0.962, and RMSEP = 0.080 mg/mL. Some *Curcuma* samples and *Saffron* samples were added to the training set and after the secondary feature wavelength screening: The Support Vector Machines (SVM) quantitative model predicted *R_p_*^2^ fluctuated in the range of 0.780 ± 0.035 for the content of AO in *Saffron* when 579, 781, 1195, 1363, 1440, 1553 and 1657 cm^−1^ were selected as characteristic wavelengths; the Partial Least Squares Regression (PLSR) model predicted *R_p_*^2^ fluctuated in the range of 0.500 ± 0.035 for the content of AO in *Curcuma* when 579, 811, 1195, 1353, 1440, 1553 and 1635 cm^−1^ were selected as the characteristic wavelengths. The robustness and generalization performance of the model were improved. (4) Conclusion: In this study, it has been discovered that the combination of surface-enhanced Raman spectroscopy (SERS) and machine learning algorithms can effectively and promptly detect the content of AO in various types of TCMs.

## 1. Introduction

In recent years, the market for Chinese herbal medicines has lacked standardization, leading to occasional events of adulteration and counterfeiting within the industry of Chinese Herbal Slices. Dyeing and adulteration are the most common phenomenon. The main coloring agents causing concern in the marketplace are pigments, food coloring, and inorganic dyes [1]. Auramine O (AO) is a contact carcinogen that causes conjunctivitis, dermatitis, and irritation of the upper respiratory tract. Prolonged exposure can lead to liver and kidney damage, as well as more severe cases of cancer and other illnesses [2]. In addition, AO should not be found in Chinese herbal medicines, proprietary Chinese medicines, or Chinese herbal medicine tablets as it is a non-food substance, according to the National Drug Administration. However, some dishonest producers still abuse AO to dye Chinese herbal medicines.

*Curcuma xanthorriza Roxb* (*Curcuma*), derived from the dried root tuber of the ginger plant *Curcuma longa* L., has various medical applications, including promoting blood circulation, relieving pain, antioxidant and anticancer [3,4,5]. *Dendrobium officinale* (*D. officinale*), sourced from the dried stem of the orchid plant *Dendrobium officinale Kimura et Migo*. As a traditional and highly valued Chinese medicinal substance, it possesses the functions of enhancing digestive health, alleviation of alcoholic fatty liver [6], improving respiratory function, alleviating coughs, combating inflammation, and reducing blood glucose levels [7]. Additionally, *Saffron* is known for its therapeutic properties as a traditional Chinese herbal medicine, including the ability to enhance blood circulation, eliminate blood clots, detoxify the body, alleviate depression, and promote relaxation of the nervous system [8,9]. Most of the existing research on the adulteration of valuable Chinese herbal medicines is focused on the identification of counterfeit products. Jana [10] chose ultra-high-performance reverse-phase liquid chromatography coupled with tandem high-resolution mass spectrometry to detect the following potential botanical adulterants used for a Saffron substitution or dilution. Similarly, Younis [11] chose the UPLC-HR-MS/MS technique to detect adulteration in five traded Saffron. Pejman [12] based on soft computing methods, Vis-Nir imaging, and chemical analysis to detect and classify Saffron adulterants.

In addition to the conventional detection techniques, novel methodologies are extensively utilized to identify fraudulent goods in Chinese herbal medicines. Holz [13] used loop-mediated isothermal amplification (LAMP) and lateral-flow-assay (LFA) to detect the *Carthamus tinctorius* and *Curcuma* longa in *Saffron*, which can achieve detection adulterants in *Saffron* within 25 min. In 2019, Yang [14] used LAMP to identify 17 samples of *D. officinale* and 32 adulterant samples from other Dendrobium species. Molecular biological methods for the detection and identification of Saffron and its adulterants are used by Bansal [15] and Zhao [16]. Normal light and fluorescence microscopy can be employed for identifying plant species. Chu [17] used these techniques to distinguish *D. officinale* from three commonly misidentified species. However, it is impossible to establish a high-performance liquid chromatography (HPLC) detection method due to the lack of resources, or the varieties with similar provenances can not be identified by microscopic observation, such as *Dendrobium huoshanense*. Hu [18] designed polymerase chain reaction-restriction fragment length polymorphism (PCR-RFLP) primers, which provided a new idea for the rapid identification of *Dendrobium huoshanense*. Also, as a part of molecular biology, Wang [19] used a DNA barcode to accurately identify *Dendrobium huoshanense* and its common adulterant *Dendrobium candidum*.

With the rapid development of spectral technology, new methods have been developed to identify counterfeit Chinese herbal medicines. Li [20] employed cloud-internet portable near-infrared spectroscopy along with chemometrics to differentiate five prevalent counterfeit products in the *Saffron*. Attenuated-total-reflectance Fourier-transform infrared (ATR-FTIR) spectroscopy and chemometrics are used to detect the expired *Saffron* in fresh *Saffron* [21]. To determine the quality of *Saffron*, Dai [22] developed a method which is a thin layer chromatography technique coupled with Raman spectroscopy, which can determine the artificial pigments (red 40 or yellow 5 at 2–10% (*w*/*w*)) and the other plant adulterants such as *Safflower* and *turmeric* at 20–100% (*w*/*w*). The multispectral system, coupled with Principal Component Analysis (PCA) and Bhattacharyya distance, gave an *R*^2^ detection rate of 0.99 for tartrazine in *turmeric* powder [23].

Although there have been many studies on the detection of counterfeit products in *Saffron*, *Curcuma*, and *D. officinale*. However, the detection and use of AO in three traditional Chinese medicines have not been described. Aftermarket research has revealed that the surface color of *Curcuma* tends to become lighter due to extended storage time or high storage temperatures, which makes it possible for criminals to dye it illegally with the AO [24]. Because of the high price of *Saffron*, illegal traders use AO to process and dye cheap daylilies and grass seedlings to make counterfeit products [25]. The glossy surface of *D. officinale* is characteristically yellow-green. Therefore, AO was usually used to dye its desired color [26].

At present, the detection methods of AO in food and medicine mainly include chromatography [2,27] and fluorescence [28,29,30], which have the advantages of high accuracy and good detection sensitivity. In addition, adsorptive stripping voltammetry [31], bionanosorbent [32], and terahertz spectroscopy [33,34] have also been applied for the determination of AO in foods and herbs. However, these types of detection methods have some disadvantages, such as expensive instruments, complicated sample processing, and complicated operation, which greatly limit their wide application in field detection. So, it is urgent to develop a rapid and simple method to detect the AO in different Chinese herbal medicines.

Raman spectroscopy (RS) is a spectroscopic technique for characterizing molecular structure based on the Raman effect, including molecular vibration and rotation [35]. Since it was discovered by Indian scientist C.V. Raman in 1928, with the continuous progress and development of computer and signal detection technology, Raman spectroscopy technology has gradually become more intelligent, non-destructive, convenient, and fast [36]. However, the fluorescence phenomenon, background noise, background baseline, and other signals will affect the weak Raman signal, which will reduce the signal-to-noise ratio of the spectrum and increase the analysis error, thus limiting its development in trace detection. Surface-enhanced Raman spectroscopy (SERS) is a technology in which the existence of metal nanoparticles can cause a plasma effect to enhance Raman spectrum signals. This technology was proved experimentally in 1977 [37]. C.V. Raman and K.S. Krishnan discovered Raman scattering, which is characterized by a small energy difference between a part of scattered light and incident light. With the rapid development of Raman spectroscopy technology, SERS was discovered and widely used. SERS is more specific and sensitive than Raman spectroscopy and can enhance the Raman signal of the molecule to be detected by 10^13^~10^15^ times. Therefore, SERS technology has a long-term application prospect in the detection of adulteration of Chinese herbal medicines [35], pesticide residues [36], and food safety testing [37].

SERS has great potential and advantages in the rapid detection of AO due to its good sensitivity, high accuracy, and simple and rapid operation. Shao [38] established a rapid qualitative and quantitative method for AO in beverages based on SERS. The minimum detection limits of AO pigment in carbonated and functional drinks are as low as 2.5 μg/L and 5.0 μg/L, which has the advantages of simple sample pretreatment, portable detection equipment, fast detection speed, and low detection cost. Yan [39] improved the detection sensitivity of basic light yellow II, basic orange II, and soap yellow in bean products by optimizing the pretreatment steps, and the detection limits were 3.0, 1.0, and 4.0 mg/kg, respectively. Zhang [40] used Dynamic Surface-enhanced Raman spectroscopy to remove the interfering impurities in the spectra acquisition process by dispersed matrix solid phase extraction and finally found that the established method can realize rapid detection of AO in Yuba, with the quantitative limit of 0.5 mg/kg.

Fewer studies are using SERS for illicit pigments in Chinese herbal medicines. Li D. [41] used paper-based surface-enhanced Raman spectroscopy to realize the rapid, simple, and non-destructive detection of four common dyes, such as AO, in adulterated *Saffron*, which met the requirements of rapid on-site detection. In the same year, Li D. [42] used AgNPs as a substrate for surface-enhanced Raman scattering, resulting in a fast, convenient, and extremely sensitive platform for dye adulteration detection in medicinal herbs, which was able to detect nine different dyes in TCMs. Zhang [25] optimized the wetting agent concentration, spraying time, and spraying amount of AgNPs and realized the rapid detection of four dyes such as AO in *Saffron* by thin-layer chromatography coupled with surface-enhanced Raman spectroscopy (TLC-SERS). Solomon [43] coupled the SERS sensor with multivariate models for the detection of Sudan II and IV in palm oil.

Most of the current research concentrates on identifying illicit coloring agents in individual Chinese herbal medicines, and there are fewer studies on identifying illegal pigments in a variety of Chinese herbal medicines. Therefore, in this study, AgNPs were used as the enhancement substrate for SERS and combined with the machine learning method for the rapid and accurate detection of AO content in Chinese herbal medicines.

In recent years, the method of SERS combined with machine learning for the determination of target substances has gained widespread use with the advancement of chemometrics. Li [44] combined SERS with a deep learning convolutional neural network (CNN) algorithm to detect the thiram and pymetrozine in tea, with correlation values of 0.995 and 0.977. The method proposed by Sha [45] is to identify midazolam and diazepam by combining SERS with CNN. The coefficient of detection reaches up to 0.966. Raman spectroscopy combined with a long short-term memory (LSTM) neural network has enabled the prediction of microplastics in olive oil [46].

The sparrow search algorithm (SSA) is a new algorithm of swarm intelligence optimization algorithm, which has the characteristics of strong optimization ability and small error. It can not only accelerate the convergence speed of the backpropagation (BP) neural network but also avoid the result falling into local extreme value [47,48]. Therefore, this technique has widespread applications in gas detection [49], trajectory prediction [50,51], and other relevant industries. Luo [47] combined SSA-BP with laser-induced breakdown spectroscopy to rapidly detect the contents of Cd, Cu, and Pb in *Fritillaria thunbergii*, with *R_p_*^2^ of 0.972, 0.991, and 0.956, respectively. Nevertheless, the integration of SERS with SSA-BP in conventional Chinese medicine has not been documented.

In general, the objectives of this study are as follows: Firstly, a prediction model for the content of AO in Chinese herbal medicines *Curcuma*, *D. officinale,* and *Saffron* was established based on SERS Raman spectroscopy combined with PLSR, SVM, and SSA-BP methods; then, the generalizability of different models among different herbs was investigated. The method established in this study provides a rapid detection method for predicting the content of adulterated pigment AO in different Chinese herbal medicines, which has the effect of achieving rapid on-site detection, curbing the fight against illegal dyeing and counterfeiting, and ensuring the safety of Chinese herbal medicines to protect people’s health.

## 2. Materials and Methods

### 2.1. Materials

AO was purchased from Shanghai Yuanye Bio-Technology Co. (Shanghai, China). Accurately weigh 10 mg of AO powder, dissolve in water, and transfer to a 10 mL constant volume volumetric flask. Using the above solution as the original solution, solutions of AO were prepared at concentrations of 0.500 mg/mL, 0.100 mg/mL, 0.050 mg/mL, 0.010 mg/mL, 0.005 mg/mL and 0.001 mg/mL. The *Curcuma* used in the experiment was produced in Leshan, Sichuan Province, and washed the surface soil section, respectively, with the above concentration gradient AO staining, each gradient staining 20 groups, drying spare. The *D. officinale* was produced in Taizhou, Zhejiang Province, China, and the *Saffron* was produced in Naqu, Tibet, China, dyed as above, and then dried.

The laboratory self-assembled Raman spectroscopy detection equipment, including QE-Pro spectrometer (Ocean Optics, FL, USA), Laser-785 nm laser, Y-type optical fiber, Raman detection probe, and object stage. The detection principle is as follows: The 785 nm laser beam is transmitted through an optical fiber to the probe, which focuses on the surface of the object and excites the Raman scattered light. The probe transmits the Raman signal to the spectrometer through the optical fiber and focuses it on the Charge coupled Device (CCD) array, which enables rapid acquisition of Raman scattering from the detected object. The schematic diagram of spectrum acquisition was shown in Appendix A.

### 2.2. Preparation and Characterization of AgNPs

AgNPs were prepared according to the Frens method [52]. Accurately weigh 1.06 mL of silver nitrate solution with a concentration of 0.100 mol/L, add ultrapure water to a constant volume of 100 mL, stir while heating under the condition of an oil bath until the solution slightly boils, and add 2 mL of 1% sodium citrate solution drop by drop, continue heating at a constant temperature for 1 h and observe that the color of the solution changes from colorless to yellow and finally to grey-green, then cool to room temperature and store in a refrigerator at 4 °C The ultraviolet absorption spectrum (SHANGHAI METASH INSTRUMENTS Co., Ltd., Shanghai, China) results of AgNPs prepared by the Frens method are shown in Figure 1a.

The preparation method for 30-fold concentrated AgNPs is to accurately weigh 30 mL of the prepared AgNPs into a centrifuge tube, centrifuge at a speed of 5000 r/min for 20 min, discard 29.5 mL of the supernatant, and add 0.5 mL of ultrapure water. The results of the transmission electron microscope (Thermo Fisher Scientific, MA, USA) are shown in Figure 1b. It can be seen from the figure that the AgNPs prepared in this study have a maximum absorption at 419 nm and are a single characteristic wavelength, so it is preliminarily judged to be spherical silver nanoparticles with good dispersibility. TEM images show that it is spherical in shape and about 50 nm in diameter.

### 2.3. SERS Spectra Acquisition

Firstly, AgNPs and AO solution were prepared in the ratio of 2:1, 3:1, and 4:1 by volume to determine the Raman characteristic peaks of the AO solution. Then, the previously stained and reserved herbal medicines were prepared and tested according to the above ratios. The sample to be detected is placed on the detection platform, and the detection is based on the Raman spectrum detection system built by the laboratory. The parameters of the Raman spectrometer are set as follows: laser power 320 mW, integration times 2 times, integration time 3000 ms, and sampling distance 2 mm. To reduce fluorescence interference, it was collected in a black box. Each sample is tested three times at different test points and then the average value is taken as the final Raman spectrum data of the final sample.

### 2.4. Sample Set Division

The 100 samples of *D. officinale* were divided into calibration and test sets in a ratio of 7:3 using the SPXY algorithm. Thirty samples of *Saffron* and *Curcuma* with different concentrations were randomly selected as the test set, and the subsequent test sets for predicting AO content in *Saffron* and *Curcuma* refer to the above samples. The remaining 55 samples of *Saffron* and 60 samples of *Curcuma* were divided into groups with 5 samples in each group starting from the low concentration of AO and then added to 70 *D. officinale* training sets according to the experimental needs. Sample information and data set division are shown in Table 1.

### 2.5. Data Pre-Processing and Characteristic Wavelength Screening

The selection of appropriate pre-processing methods for the collected Raman spectra can not only effectively eliminate the background noise, but also optimize the spectra data and improve the prediction ability of quantitative models. In this study, the selected spectra range is 400–1800 cm^−1^, and the spectra pre-processing methods include SG smoothing, airPLS baseline correction, standard normal variable transformation (SNV), multivariate scattering correction (MSC), and their combinations [53].

After removing the edge bands with a low signal-to-noise ratio of the spectra data in the spectra curves, the Competitive Adaptive Re-weighting Algorithm (CARS) was adopted to select the characteristic wavelengths, which was used to improve the speed of model computation and increase the prediction accuracy [54]. The spectra data of 100 *D. officinale* samples stained with different concentrations of AO were screened by CARS to determine nine characteristic wavelengths such as 452, 551, 649, 778, 1195, 1353, 1444, 1553, and 1635 cm^−1^.

### 2.6. Method

#### 2.6.1. PLSR

In this study, the PLSR model was used to predict the content of AO in different Chinese herbal medicines. This approach amalgamates the benefits of MLR and PCA. It is better suited for solving the regression problem with more bands and greater autocorrelation than the traditional linear regression method [55]. Firstly, the PLSR diminishes the dimensions of the multivariate variables Y and X and transforms the original variables into orthogonal principal components. This extraction not only identifies the principal components in X and Y but also maximizes the correlation between the extracted principal components, thus eliminating spectral overlap information.

In this experiment, spectral data were designated as the independent variable X, while the concentration of AO was designated as the dependent variable Y. To avoid the overfitting of the model, the cross-validation strategy was used to determine the number of latent variables in PLSR. The final result is the comparison between the predicted value of the model and the real value. The PLSR model performance is evaluated using *R*^2^ and RMSE.

#### 2.6.2. SVM

The SVM is a classical supervised machine learning algorithm, that can effectively deal with various nonlinear problems brought by small samples and high-dimensional data, so it has been widely used in many fields. The algorithm’s principle involves mapping sample data into a high-dimensional space and converting nonlinear problems into linear ones. Nevertheless, regression calculations after elevation mapping require more computation as the feature dimension of the sample data increases. To tackle this problem, a kernel function has been added to the SVM [22,56]. The kernel function avoids the problem of high computational complexity, which enables SVM to deal with complex data sets and improves the generalization ability of the model. This study employs the Radial Basis Function (RBF) kernel function. Similarly, to avoid over-fitting of the model, the penalty factor *c* and insensitive coefficient *g* in the SVM kernel function were optimized by using the cross-validation strategy, which improves the accuracy of prediction results.

#### 2.6.3. SSA-BP

The SSA is a group-based optimization technique, which was introduced by Xue Jiankai et al. in 2020. Its inspiration is mainly derived from the foraging behavior of sparrows. The algorithm categorizes the sparrow population into three groups: discoverers, joiners, and scouts. By considering their predatory behaviors, the search for targets can be optimized [47]. It has the characteristics of simple results, few control parameters, and strong local search ability.

The BP neural network is a commonly used machine learning algorithm. However, it is easy to fall into the local optimal solution in the training process, which leads to the poor prediction performance of the model. Therefore, it is necessary to introduce the SSA algorithm to optimize it [51]. By updating the positions of discoverers, joiners, and scouts in turn, the global optimal threshold and weight are finally output. Then input the optimized threshold and weight into the BP neural network to train and predict the data [57]. Using spectral data as the input factors, prediction models were established to detect AO in various Chinese herbal medicines. The schematic of the SSA-optimized BP neural network is shown in Figure 2. In addition, the Layer-wise Normalization was used, resulting in a smoother optimization landscape for neural network loss.

#### 2.6.4. Models Building Procedure

It is important to select a suitable model for predicting the AO content of adulterated pigments in experimental samples. In this study, we compared the effectiveness of Partial PLSR, SVM, and SSA- BP SSA-BP for predicting adulterant pigment AO content in herbal medicines, and selected the method with higher measurement accuracy to establish the adulterant pigment detection model. *D. officinale* was used as the source sample, and *Saffron* and *Curcuma* were used as the target samples. The 30 of 85 *Saffron* and 90 *Curcuma* were randomly selected as the test set (including each concentration of AO). The experimental study was carried out from three aspects. First, PLSR, SVM and SSA-BP calibration models were developed for 70 stained *D. officinale* set samples for the content of AO in Chinese herbal medicines, and 30 stained *D. officinale* test sets were used to predict the accuracy of the models; then, 70 *D. officinale* training samples, PLSR, SVM and SSA-BP calibration models were established, respectively, samples, and the contents of AO in 30 test concentrations of *Saffron* and *Curcuma* were predicted; finally, the remaining *Saffron* and *Curcuma* samples were sequentially grouped from low to high concentration, with each five as a group, and five *Saffron* samples and five *Curcuma* samples were added each time to the training samples consisting of 70 *D. officinale* samples, and the new sample set was used as a correction set to establish correction model. The regression model was used to predict the AO content in 30 *Saffron* samples and *Curcuma* samples until all samples were added. Comparison of the prediction accuracy of the *D. officinale* calibration model for the content of AO in *D. officinale* and two other Chinese herbal medicines; analysis of the relationship between the number of target samples added to the training set and the performance of the model. The spectral preprocessing method, modeling method and formula used in this study were shown in Table 2.

### 2.7. Model Evaluation Indexes

The determination coefficient (coefficient of determination, $R^{2}$) and the root mean square error of prediction (RMSEP) were used as model evaluation indicators.
$$R^{2}=1-\frac{\sum_{i=1}^{n}{\left(y_{i}-{\hat{y}}_{i}\right)}^{2}}{\sum_{i=1}^{n}{\left(y_{i}-{\bar{y}}_{i}\right)}^{2}}\;RMSEP=\sqrt{\frac{1}{n-1}\sum_{i=1}^{n}{\left(y_{i}-{\hat{y}}_{i}\right)}^{2}}$$

In the above formula, n is the number of samples in the test set; $y_{i}$ is the true value of AO content in the i sample in the test set; ${\hat{y}}_{i}$ is the predicted value of AO content in the i sample in the test set; ${\bar{y}}_{i}$ is the average value of AO content in the test set samples. The prediction decision coefficient $R^{2}$ is used to measure the correlation between the predicted value and the real value, and the closer it is to 1, the higher the correlation of the model is. RMSEP is used to measure the error between the predicted value and the true value of AO content in the test set samples, and the closer to 0, the stronger the prediction ability of the model [60].

The technical flow chart of this study was shown in Figure 3.

## 3. Results and Discussion

### 3.1. SERS Spectra Analysis of AO

The Raman spectra intensity of AO solution mixed with concentrated AgNPs in different ratios are shown in Figure 4a. The AO solution without AgNPs has no obvious characteristic Raman wavelength. When the volume ratio of AgNPs to AO solution is 2:1, noticeable Raman signals appear in 1000–1700 cm^−1^; while in 400–1000 cm^−1^, there is no effective Raman characteristic wavelength. It can be seen that the addition of AgNPs has a certain effect on the enhancement of the Raman signal in the AO solution. The reason might be that AO is a cationic molecule, while the AgNPs also carry a positive charge, the two molecules rely on physical adsorption to achieve the interaction. there are fewer AgNPs in the system and the enhancement effect is unsatisfied. When the volume ratio is 3:1, the Raman wavelength has been significantly enhanced in the whole range of 400–1800 cm^−1^. That may be due to the increase in the content of AgNPs in the system and the increase in the contact area of the AO molecules. However, when the proportion of AgNPs in the system was further increased and the volume ratio was adjusted to 4:1, the concentration of AO solution was diluted and the Raman characteristic wavelength intensity decreased compared to that at 3:1. It suggests that the Raman enhancement effect is unsatis factory due to inappropriate dosage of AgNPs.

The Raman characteristic wavelength of AO standard solution at a 3:1 volume ratio of AgNPs and AO are shown in Figure 4b and Table 3. According to the relevant literature [40], the intense bands at 551 cm^−1^ and 649 cm^−1^ are attributed to υ(C-H). The band at 735 cm^−1^ is assigned to δ(H-C-H); the intense band at 778 cm^−1^ is attributed to δ(C-N-C). The band at 1189 cm^−1^ is assigned to υ(C-C), and the bands at 1438 cm^−1^ and 1481 cm^−1^ are attributed to ring stretching vibrations. The intense band at 1598 cm^−1^ is attributed to τ(C-N). The Raman band at 778 cm^−1^ is more clearly visible in the low-concentration solution so it is identified as the characteristic wavelength of AO in the subsequent study [25,41,61].

The SERS signals were collected from the *Curcuma*, *D. officinale*, and *Saffron*, which were dyed and prepared in advance, dropping the concentrated AgNPs in different ratios. Taking the standard characteristic wavelength at 778 cm^−1^ as a reference, it was observed that the SERS band was strongest when the volume ratio of AgNPs to AO was 3:1, as shown in Figure 4c. The SERS signals of the three herbal medicines were measured at a volume ratio of 3:1. The analysis of the detection limit of AO solution in three kinds of Chinese herbal medicines was carried out according to the above conditions, and the results are shown in Figure 5. When the concentration of AO in the dye solution was as low as 0.010 mg/mL, the color of the dye solution was no longer visible at this time, but the characteristic wavelength at 778 cm^−1^ was still clearly visible during the SERS detection in the three kinds of Chinese herbal medicines. According to the principle of three times noise [62], the LOD and LOQ of the method established in this study are 0.003 mg/mL and 0.009 mg/mL respectively. Thus, this method can be used for the trace detection of AO dyes in three kinds of Chinese herbal medicines, given the prohibition of AO in Chinese herbal medicines. With the decrease in AO dye concentration, the SERS signal decreased gradually. 

### 3.2. Model Establishment and Research

#### 3.2.1. Predictive Modelling of AO Content in *D. officinale*

The study established PLSR, SVM, and SSA-BP regression models for the adulterant pigment AO in *D. officinale*. The results are shown in Figure 6 and Table 4. The prediction coefficients of determination of the three models were 0.940, 0.930, and 0.962, and the RMSEP were 0.099 mg/mL, 0.209 mg/mL, and 0.080 mg/mL, respectively. The results of wavelength selection by the CARS algorithm showed that the characteristic wavelengths of AO in *D. officinale* were mainly located at 452, 551, 649, 778, 1195, 1353, 1444, 1553, and 1635 cm^−1^, which corresponded to the characteristic wavelength of the AO standard. The three prediction models established can effectively predict the content of the adulterated pigment AO in *D. officinale*, and the SSA-BP model has the highest prediction accuracy. The fitness curve of SSA-BP model was shown in Figure 7.

#### 3.2.2. Generalization of the Prediction Model across Multiple Chinese Herbs

The model calibrated by *D. officinale* samples was explored to predict the AO content in different Chinese herbal medicines, *Saffron* and *Curcuma*. The results are presented in Table 5 and Figure 8. As indicated in Table 5, the performance of the PLSR model for the prediction of AO content in *Saffron* achieved *R_p_*^2^ = 0.751 and RMSEP = 0.344 mg/mL. The PLSR model predicted the content of AO in *Curcuma* with *R_p_*^2^ = 0.399 and RMSEP = 0.623 mg/mL. This suggests that the calibration model established by *D. officinale* samples is capable of predicting the AO content in *Saffron*, but not accurately predicting AO content in *Curcuma*.

#### 3.2.3. Improvement of the Prediction Precision of AO Model for Multiple Chinese Herbs

To predict the AO content of *Saffron* and *Curcuma* test samples accurately, it is necessary to prevent the model from learning the intrinsic information of Chinese medicinal materials except for the AO substance. It is proposed that the calibration sample set of the AO model could consist of more categories of Chinese medicinal materials. A new training set was rebuilt by adding *Saffron* and *Curcuma* samples into *D. officinale* samples. Five samples of *Saffron* and *Curcuma* were incremented, and the prediction effects of the three models were evaluated in Table 6. The results showed that SVM was the most effective model for predicting AO in *Saffron*, whereas PLSR was the best model for *Curcuma* among the compared models. Therefore, SVM and PLSR models were used to predict the AO content in *Saffron* and *Curcuma* in the following steps.

The results of the *R_p_*^2^ and RMSEP of the optimized model for predicting AO content in the *Saffron* and *Curcuma* test set are shown in Figure 9. As the number of *Saffron* and *Curcuma* samples added into the training set increased, the predicted *R_p_*^2^ of the calibration model gradually raised and RMSEP gradually decreased. Nevertheless, the predicted *R_p_*^2^ shows a saturation trend, requiring model stability investigation in the future.

The characteristic wavelengths were obtained by the CARS algorithm. The characteristic wavelengths of AO in *Saffron* and *Curcuma* samples were selected and compared with the 9 characteristic wavelengths of *D. officinale* previously discussed. The optimal characteristic wavelengths of *Saffron* were 579, 781, 1195, 1363, 1440, 1553, and 1657 cm^−1^; and those of *Curcuma* were 579, 811, 1195, 1353, 1440, 1553, and 1635 cm^−1^. Due to the complex composition of traditional Chinese medicine, the Raman characteristic wavelength of the adulterant pigment AO added to different herbs will be shifted. The characteristic wavelengths were chosen to build predictive models for 30 samples of *Saffron* and *Curcuma* using SVM and PLSR. Table 7 shows the changes in prediction accuracy as the number of target samples increases. Following the wavelength selection method and adding varying amounts of other herbs, the SVM and PLSR models predicted that *R_p_*^2^ fluctuated between 0.780 ± 0.035 and 0.500 ± 0.035. Compared with the prediction model calibrated by D. officinale samples, the addition of some other herbs and further feature wavelength selection, not only reduces the influence of redundant bands, and effectively enhances the accuracy and stability of feature wavelength variable selection but also simplifies the model and improves the prediction accuracy. It also provides a fast and simple prediction method for the detection of adulterated pigment AO content in different kinds of Chinese herbal medicines.

## 4. Conclusions

In this study, Surface-enhanced Raman spectroscopy and three machine learning models (PLSR, SVM, and SSA-BP) were used to explore the possibility of the chemical dye AO across multiple Chinese herbal medicines. The prediction of AO content in various Chinese herbal medicines by using *D. officinale* as a calibration model was investigated. The results showed that the SSA-BP model had the highest accuracy in predicting the content of AO in *D. officinale* with *R_p_*^2^ = 0.962 and RMSEP = 0.080 mg/mL. The PLSR model calibrated by *D. officinale* samples had acceptable prediction effects of AO content in *Saffron*, but could not effectively predict AO content in *Curcuma*. After adding some *Saffron* and *Curcuma* samples to the calibration set of *D. officinale*, the prediction accuracy of the PLSR model for AO content in *Saffron* and *Curcuma* gradually increased. As the characteristic wavelengths of *Saffron* and *Curcuma* herbs were further selected, the model changes tended to stabilize. using selected characteristic wavelengths of 579, 781, 1195, 1363, 1440, 1553, and 1657 cm^−1^, the *R_p_*^2^ of the SVM quantitative model for predicting AO content in *Saffron* varied between 0.780 ± 0.035. For the selected wavelengths of 579, 811, 1195, 1353, 1440, 1553, and 1635 cm^−1^, the PLSR model predicted *R_p_*^2^ fluctuations within 0.500 ± 0.035 for the AO content in *Curcuma*. The *D. officinale* calibration set successfully predicted the AO content in various types of Chinese herbal medicines. In this study, various model prediction methods for the content of AO in different kinds of adulterated Chinese herbal medicines were investigated and a new rapid detection method based on SERS Raman spectroscopy was proposed, which is practical to provide a detection method with higher accuracy, reduced model calculation and lower detection cost for the detection of adulteration content of artificial pigment AO in Chinese herbal medicines. In the future, to enhance the reliability and effectiveness of research, it can be approached from the following two angles. On the one hand, it is necessary to explore a new SERS substrate that was based on the existing experiments and combined with flexible materials to reduce the detection limit of AO; on the other hand, expanding the representative data set which can expand the universality and diversity of this study.

## Figures and Tables

**Figure 1 foods-12-04124-f001:**
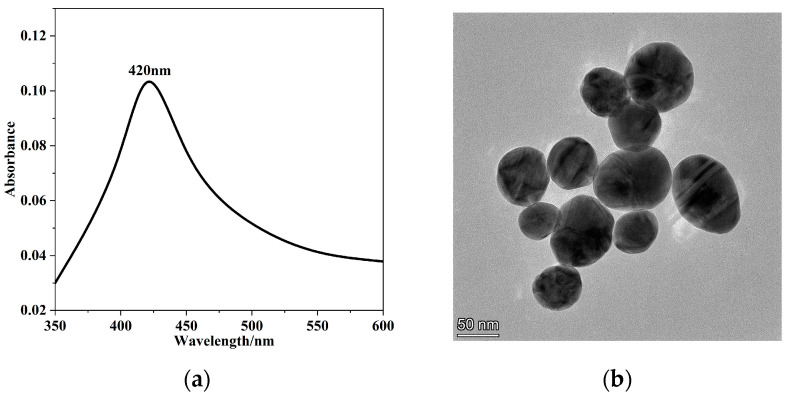
Structural characterization diagram of AgNPs: (**a**) Ultraviolet absorption spectrum of AgNPs; (**b**) Transmission electron microscope diagram of AgNPs.

**Figure 2 foods-12-04124-f002:**
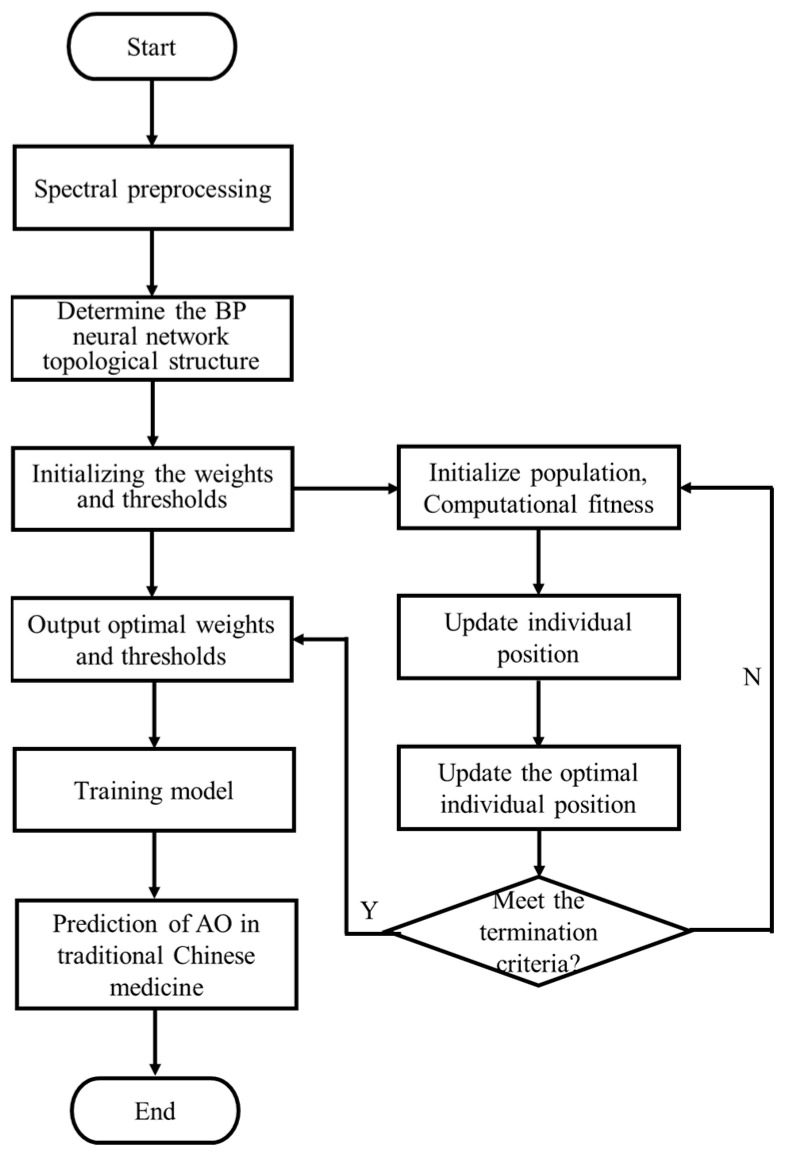
Flow chart of SSA-BP neural network regression prediction [58,59].

**Figure 3 foods-12-04124-f003:**
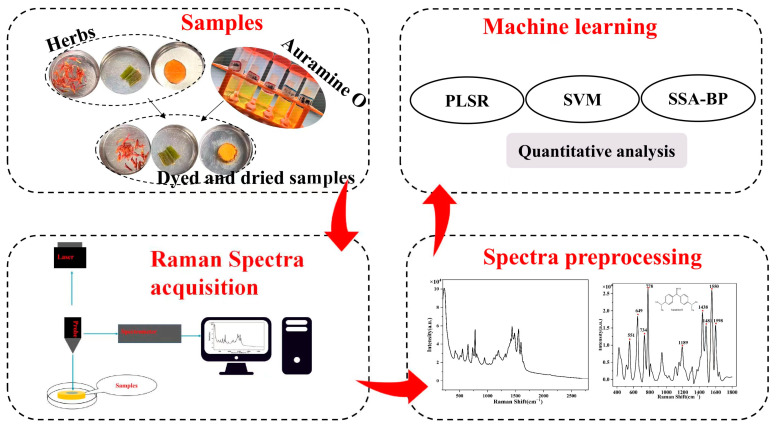
Experimental flow chart.

**Figure 4 foods-12-04124-f004:**
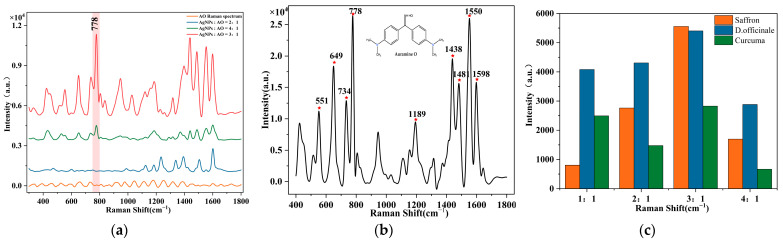
Raman spectra of Auramine O solution: (**a**) Raman spectra of Auramine O solution and AgNPs in different proportions; (**b**) Raman characteristic wavelength of Auramine O solution at 3:1; (**c**) Raman spectra at 778 cm^−1^ at different ratios of AgNPs and Auramine O solution in three Chinese herbal medicines.

**Figure 5 foods-12-04124-f005:**
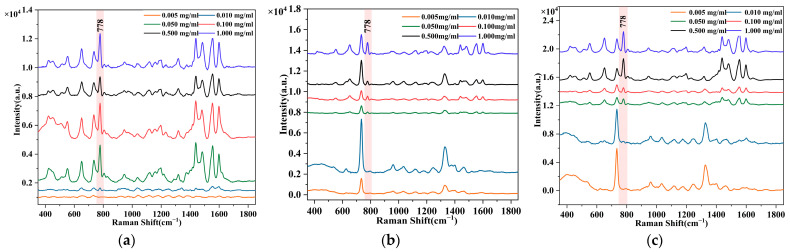
Raman spectrum of different Chinese medicines with different concentrations of AO: (**a**) Raman spectra of *D. officinale*; (**b**) Raman spectra of *Curcuma*; (**c**) Raman spectra of *Saffron*.

**Figure 6 foods-12-04124-f006:**
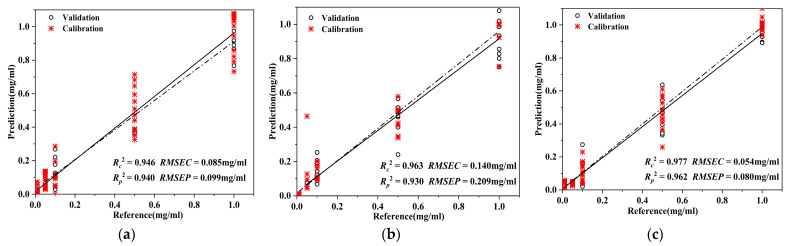
Prediction results of Auramine O content in *D. officinale*: (**a**) PLSR prediction results; (**b**) SVM prediction results; (**c**) SSA-BP prediction results.

**Figure 7 foods-12-04124-f007:**
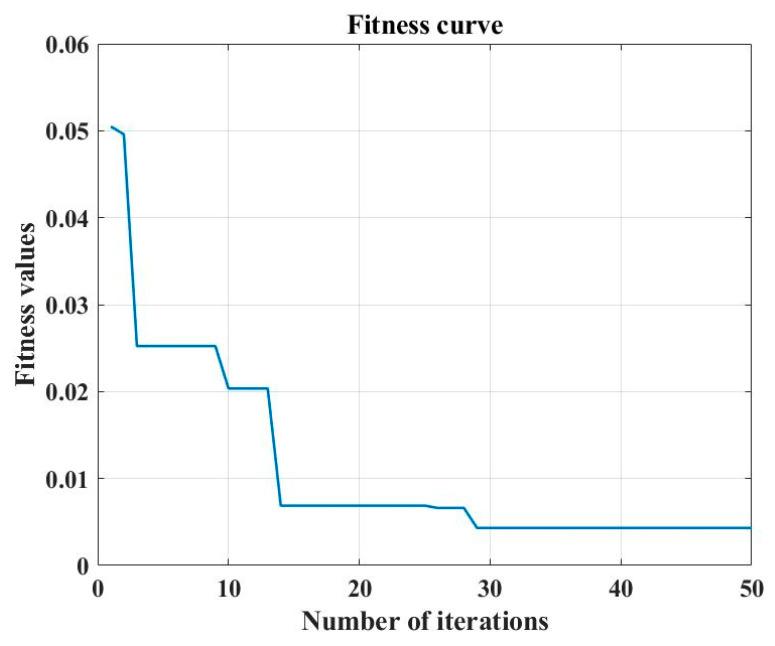
Fitness curve of SSA-BP model.

**Figure 8 foods-12-04124-f008:**
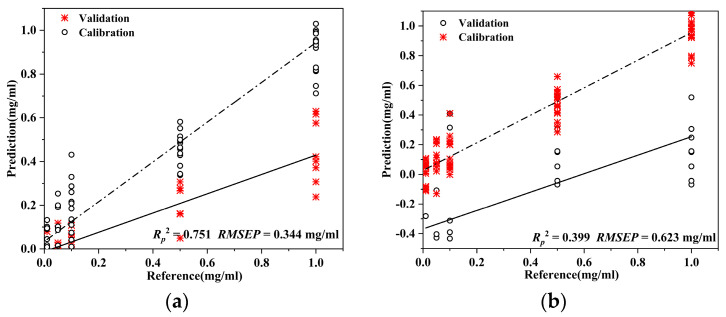
Prediction results of Auramine O content in *Saffron* and *Curcuma* using PLSR model calibrated by *D. officinale* samples: (**a**) Prediction results of Auramine O content in *Saffron* by PLSR; (**b**) Prediction results of Auramine O content in *Curcuma* by PLSR.

**Figure 9 foods-12-04124-f009:**
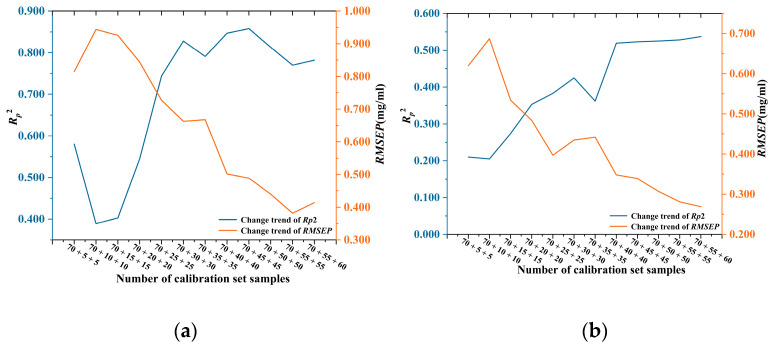
Changes of model performance after adding different number of target samples: (**a**) Variation of model performance of SVM for the prediction of Auramine O content in *Saffron*; (**b**) Variation of model performance of PLSR for the prediction of Auramine O content in *Curcuma*.

**Table 1 foods-12-04124-t001:** Sample information and data set division table.

Name	Origin	Dataset Size	Calibration Set	Test Set
*Dendrobium officinale*	Taizhou, Zhejiang	100	70	30
*Saffron*	Naqu, Tibet	85	*n* × 5 (*n* = 1, 2, …, 11)	30
*Curcuma xanthorriza Roxb*	Leshan, Sichuan	90	*m* × 5 (*m* = 1, 2, …, 12)	30

**Table 2 foods-12-04124-t002:** Spectral transformation method, modeling method and formula.

Name	Formula	Descriptions
airPLS	$F=\sum_{i=1}^{m}{\left(x_{i}-z_{i}\right)}^{2},\;\;\;\;R=\sum_{i=2}^{m}{\left(z_{i}-z_{i-1}\right)}^{2}$, Q=F+λR $Q^{t}=\sum_{i=1}^{m}\omega_{i}^{t}{\left|x_{i}-z_{i}^{t}\right|}^{2}+\lambda \sum_{j=2}^{m}\omega_{i}^{t}{\left|z_{j}^{t}-z_{j-1}^{t}\right|}^{2}$ $\omega_{i}^{t}=\left\{\begin{array}{c} 0\;\;\;\;\;\;\;\;\;\;\;\;\;\;\;\;\;\;\;\;\;\;\;\;\;\;\;\;\;\;\;\;\;\;\;\;\;\;\;\;\;\;\;\;\;\;x_{j}\geq z_{i}^{t-1}\;\;\;\;\;\\ \exp\left(\frac{t\left(x_{i}-z_{i}^{t-1}\right)}{\left|d^{t}\right|}\right)\;\;\;\;\;\;\;x_{j}<z_{i}^{t-1} \end{array}\;\right.\;\;\;\;\;\;\;$	x is the original spectra, z is the fitted spectra, m is the abscissa of spectral data, F is fidelity, R is the roughness of the z, Q is the balance parameters, $Q^{t}\;$ is the weighted penalty least square formula (t is the number of iterations)
SG	$\hat{y_{j}}=\frac{\sum_{i=-m}^{m}a_{i}x_{j+i}+a_{0}}{n}$	$\hat{y_{j}}$ is the smoothed spectra, $x_{j+i}$ is the original spectra, $a_{i}$ and $a_{0}$ is the smoothing coefficient, n is the number of data in the sliding window, m is the window width, n=2m+1.
MSC	$A=\frac{1}{n}\sum_{i=1}^{n}A_{i}$, $A_{i}=k_{i}\bar{A}+b_{i}$, $A_{i(MSC)}=\frac{A_{i}-b_{i}}{k_{i}}$	$\bar{A}$ is the average spectra, n is the spectral number, Ai is the original spectra information of article i, k is the regression coefficient, *b* is the regression constant, Amsc is the corrected spectra.
SNV	$X_{SNV}=\frac{x_{k}-\bar{x}}{\sqrt{\frac{\sum_{i=k}^{m}{\left(x_{k}-\bar{x}\right)}^{2}}{\left(m-1\right)}}}$	$X_{SNV}\;$ is the transformed spectra, $\bar{x}$ is the average value of the spectra of the i, k = 1, 2, …, m, i = 1, 2, …, n is the number of the samples.
PLSR	$X=DP^{T}+E$, $Y=UQ^{T}+F$, $U=D{\left(D^{T}D\right)}^{-1}D^{T}U$, $Y_{pre}=D_{pre}BQ$	X is the Raman spectra matrix, Y is the AO matrix, D and U are Principal factor score matrix, P is the load matrix, E and F are the fitting residual matrix.
SVM	$f\left(x\right)=\omega^{*}\phi \left(x\right)+b=\sum_{i=1}^{m}\left(a_{i}-a_{i}^{*}\right)K\left(x_{i},\;y_{j}\right)+b$	$\omega^{*}$ is the weight vector, $\phi \left(x\right)$ is the SERS spectrum, $a_{i}\;and\;a_{i}^{*}\;$ is the Lagrange factor, $K\left(x_{i},\;y_{j}\right)$ is the kernel function.

**Table 3 foods-12-04124-t003:** Bands assignment of Raman Auramine O spectra.

Wavelength (cm^−1^)	Assignment
551 cm^−1^, 649 cm^−1^	C-H stretching vibration
735 cm^−1^	H-C-H deformation vibration
778 cm^−1^	C-N-C deformation vibration
1189 cm^−1^	C-C stretching vibration
1438 cm^−1^, 1481 cm^−1^	Benzene ring stretching vibration
1598 cm^−1^	C-N twisting vibration

**Table 4 foods-12-04124-t004:** Prediction results of Auramine O content in *D. officinale* by PLSR, SVM, SSA-BP.

	Calibration Set	Test Set	Pre-Processing	*R_c_* ^2^	RMSEC/mg/mL	*R_p_* ^2^	RMSEP/mg/mL
PLSR	70	30	airPLS + SG + MSC	0.946	0.085	0.940	0.099
SVM	70	30	airPLS + SG + SNV	0.963	0.140	0.930	0.209
SSA-BP	70	30	airPLS + SNV	0.977	0.054	0.962	0.080

**Table 5 foods-12-04124-t005:** Prediction results of Auramine O content in *Saffron* and *Curcuma* by three models with *D. officinale* as calibration set.

	Calibration Set	Test Set	Pre-Processing	*R_c_* ^2^	RMSEC/mg/mL	*R_p_* ^2^	RMSEP/mg/mL
PLS	70 *D. officinale*	30 *Saffron*	SG + MSC	0.910	0.119	0.751	0.344
SVM			airPLS + SG	0.919	0.241	0.714	0.662
SSA-BP			airPLS + SG + MSC	0.981	0.054	0.628	0.238
PLS	70 *D. officinale*	30 *Curcuma*	SNV	0.929	0.106	0.399	0.623
SVM			SG + SNV	0.937	0.202	0.229	1.257
SSA-BP			airPLS + SNV	0.938	0.099	0.005	0.460

**Table 6 foods-12-04124-t006:** Prediction results of Auramine O content in *Saffron* and *Curcuma* chrysalis by different models after adding target Chinese herbal medicines.

	Calibration Set	Test Set	Pre-Processing	*R_c_* ^2^	RMSEC/mg/mL	*R_p_* ^2^	RMSEP/mg/mL
PLS	70 *D. officinale* + 5 *Saffron* + 5 *Curcuma*	30 *Saffron*	SG + SNV + MSC	0.811	0.169	0.394	0.353
SVM			SNV	0.901	0.248	0.580	0.815
SSA-BP			SG + SNV	0.879	0.135	0.224	0.344
PLS	70 *D. officinale* + 5 *Saffron* + 5 *Curcuma*	30 *Curcuma*	SNV	0.894	0.127	0.210	0.620
SVM			SNV	0.908	0.246	0.037	1.018
SSA-BP			SG + SNV	0.8323	0.159	0.005	0.999

**Table 7 foods-12-04124-t007:** Changes of model performance after adding different numbers of *Saffron* and *Curcuma* samples after secondary feature screening.

Calibration Set Sample Size	*Saffron*		*Curcuma*	
	*R_p_* ^2^	RMSEP/mg/mL	*R_p_* ^2^	RMSEP/mg/mL
70 + 5 + 5	0.751	0.549	0.494	0.428
70 + 10 + 10	0.762	0.580	0.474	0.421
70 + 15 + 15	0.792	0.592	0.468	0.297
70 + 20 + 20	0.754	0.604	0.493	0.289
70 + 25 + 25	0.754	0.727	0.495	0.305
70 + 30 + 30	0.790	0.541	0.495	0.354
70 + 35 + 35	0.791	0.668	0.477	0.384
70 + 40 + 40	0.811	0.483	0.470	0.337
70 + 45 + 45	0.814	0.447	0.476	0.313
70 + 50 + 50	0.807	0.411	0.493	0.299
70 + 55 + 55	0.812	0.341	0.510	0.288
70 + 55 + 60	0.782	0.414	0.525	0.283

## Data Availability

The data used to support the findings of this study can be made available by the corresponding author upon request.

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
