# Peer review of "Rapid Indentification of Auramine O Dyeing Adulteration in Dendrobium officinale, Saffron and Curcuma by SERS Raman Spectroscopy Combined with SSA-BP Neural Networks Model"

_foods, 2023, doi:10.3390/foods12224124_

Round 1

Reviewer 1 Report

Comments and Suggestions for Authors

This article provides information on the use of surface-enhanced Raman spectroscopy (SERS) and neural networks for the rapid identification of Auramine O dyeing adulteration in traditional Chinese medicines. It includes details on the experimental setup, data analysis, and results. Additionally, the article discusses the potential applications of this approach in the quality control of traditional Chinese medicines and the future directions for research in this area.

The highlights of this article are:

- The use of SERS Raman spectroscopy and neural networks for the rapid identification of Auramine O dyeing adulteration in traditional Chinese medicines.

- The development of a novel approach that combines SERS Raman spectroscopy with SSA-BP neural networks to improve the accuracy of detecting Auramine O.

- The successful application of this approach to Dendrobium officinale, Saffron, and Curcuma, which are commonly used in traditional Chinese medicines.

- The potential of this approach for the quality control of traditional Chinese medicines and the detection of other types of chemical dyes.

We expect the authors to provide appropriate and convincing answers to the following questions:

  1. What is the significance of detecting Auramine O in traditional Chinese medicines?
  2. How does the SSA-BP neural network model improve the accuracy of detecting Auramine O?
  3. Can this approach be applied to other types of chemical dyes in traditional Chinese medicines?
  4. Are the sample sizes used in this study sufficient to draw meaningful conclusions?
  5. How does this approach compare to other existing methods for detecting chemical dyes in traditional Chinese medicines?
  6. Are there any limitations or challenges associated with the use of SERS Raman spectroscopy and neural networks in this context?
  7. How does the SSA-BP neural network used in this study compare to other types of neural networks in terms of accuracy and efficiency?

The following questions about each section of the article will be addressed.

1.      Introduction: I want to highlight that your literature review is currently insufficient. In the revised version, it is crucial to provide a more comprehensive analysis of the existing literature, particularly focusing on counterfeiting in products like medicinal plants such as saffron. Recent publications in this area need to be thoroughly addressed to enrich the discussion. Additionally, consider discussing the application of machine learning methods, especially neural networks, which have become increasingly relevant in this field.

2. Materials and Methods:

-  Your explanation of the utilized models lacks detail, leading to confusion among readers. Figure 3 represents a general neural network process, not specifically aligned with the outlined study. Adding more comprehensive details to this section would enhance clarity.

- It is recommended to create a table providing comprehensive information on the dataset size, sample specifications, and divisions to offer readers a more in-depth understanding.

- Consider providing a detailed and specific explanation for each of the PLSR, SVM, and SSA-BP regression models in a dedicated subsection within the materials and methods section to ensure clarity and thoroughness.

3. Results and Discussion

Your results and discussion are generally acceptable, but certain revisions are necessary for further improvement:
1. It is crucial to present the prediction results of the three models across the training, testing, and overall stages in a tabular format.
2. Review the size of the test training set to ensure accuracy.
3. Provide a comprehensive table outlining the statistical properties and the methodology for converting SERS spectra into model inputs.
4. Consider addressing the limitation of the limited range of OA values in your work, and ensure a more diverse and extensive representation of OA levels in your samples for improved validity and reliability of results.

Reviewer 2 Report

Comments and Suggestions for Authors

The paper deals with the investigation and selection of conditions for SERS use for the determination of trace amounts of active compounds of medical herbs using Auramine O; along with the use of machine learning. The paper is planned and presented quite clearly and, in my opinion, is free from basic errors.

However, some minor issues should be taken into account:

- Authors should be more careful with terms. Throughout the text the same parameter, Raman shift is called wavenumber or wavelength.

- Apart from re-checking the generic issues of the English, the authors should carefully check the terminology like the incorrect use: "dilution... of concentration", "spectral" and "spectra", etc,

- Figure 2a. The spectrum is of little use without the concentration value

- Figure 6. There is no LOD in these spectra, these are spectra, from which LODs are calculated, the figure shows no sign of such calculations, just a visual aid to show, where these LODs are. The caption should be rephrased.

- Figure 5. It is a bit clearer than Figure 6, but I recommend rephrasing the text as well, there is no 'analysis' in these spectra.

- The number of significant digits in error values should be kept at 1 (a single significant digit). Limits of detection are 100% error, the use of two digits for these values should be avoided.

- The authors propose the method for quantitative analysis, although they operate with LODs, the parameters of qualitative analysis. I suggest adding the limits of quantification.

- Figure 1 is too simple and can be easily omitted or moved to the Supplementary; moreover, its smaller version is shown in Figure 4

- The historical introduction to Raman spectroscopy is not needed, this is a common fact, as well as Sir K.S. Krishnan should also be mentioned anyway as a co-discoverer. I suggest removing this part (lines 66-80) almost completely as obvious and start already with SERS possibilities in modern analytical/applied spectroscopy.

Comments on the Quality of English Language

The authors should double-check the text. There are many misspelling, missed spaces (especially, before references), plural/singular use, capitalization, etc.

Round 2

Reviewer 1 Report

Comments and Suggestions for Authors

Dear Editor,

I trust this message finds you well. I apologize for any confusion, and I appreciate your understanding. I had indeed reviewed the files previously, but due to an unforeseen issue, they were inadvertently removed from my checklist, and I was unable to convey my assessment.

First and foremost, I would like to commend the authors for their diligence in addressing the review comments and making the necessary revisions. I have carefully reviewed the responses and corrections, and I am satisfied with the improvements made.

However, there is a critical concern that I believe warrants further clarification. It pertains to the limited number of surfaces used for the calculation of Auramine O, as evident in Figure 6 and Figure 8. Specifically, there are only a small number of levels for Auramine O changes. In such a scenario, the application of a neural network for modeling raises questions. When the input function exhibits minimal variation, the resulting model may struggle to produce a consistent and smooth response, even if the prediction errors are minimal. I believe it is imperative that the authors provide a satisfactory explanation for this issue before proceeding with publication.

I kindly request the authors to address this concern to ensure the robustness of their work and to provide clarity on the suitability of the neural network in these circumstances.

Thank you for your attention to this matter.

Sincerely,
